



# Development of a numerical workflow based on μ-CT-imaging for the determination of capillary pressure-saturation-specific interfacial area relationship in two-phase flow pore-scale porous media systems: A case study on Heletz sandstone

Aaron Peche[1,2], Matthias Halisch[3], Alexandru Bogdan Tatomir[2]

[1] Leibniz-Universität Hannover, Institute of Fluid Mechanics and Environmental Physics, Appelstraße 9A, 30167 Hannover, Germany

[2] Georg-August Universität Göttingen, Department of Applied Geology, Goldschmidtstraße 3, 37077 Göttingen, Germany

[3] Leibniz Institute for Applied Geophysics, Department 5, Petrophysics and Borehole Geophysics, Stilleweg 2, D-30655 Hannover, Germany

*Correspondence to*: Aaron Peche (peche@hydromech.uni-hannover.com)

**Keywords:** Two-phase Flow, CO2, Interfacial Area Surface, FEM- based Pore Scale Modeling, μ-CT

**Abstract.** In this case study, we present the implementation of a FEM-based numerical pore-scale model that
enables to track and quantify the propagating fluid-fluid interfacial area on highly complex μ-CT obtained geometries. Special focus is drawn to the reservoir specific capillary pressure ($p_c$)- wetting phase saturation ($S_w$)- interfacial area ($a_{wn}$)- relationship. The basis of this approach are high resolution μ-CT images representing the geometrical characteristics of a georeservoir sample. The successfully validated two-phase flow model is based on the Navier-Stokes equations, including the surface tension force in order to consider capillary effects for the
computation of flow and the phase field method for the emulation of a sharp fluid-fluid interface.

In combination with specialized software packages, a complex high resolution modeling domain could be obtained. A numerical workflow based on REV-scale pore size distributions is introduced. This workflow aims at the successive modification of model and model setup for simulating such a type of two-phase problem on asymmetric
μ-CT-based model domains. The geometrical complexity is gradually increased starting from idealized pore geometries until complex μ-CT-based pore network domains, whereas all domains represent geostatistics of the REV-scale core sample pore size distribution. Finally, the model could be applied on a complex μ-CT-based model domain and the $p_c$-$S_w$-$a_{wn}$ relationship could be computed.





## 1 Introduction

Understanding the evolution of the fluid-fluid interfaces in two-phase flow in porous media systems is relevant for a series of engineering and technological applications (e.g., CCS, nuclear waste repository, oil recovery, etc.) (Hassanizadeh and Gray, 1990; Joekar-Niasar et al., 2008; Niessner and Hassanizadeh, 2008; Reeves and Celia,

1996). Several experimental and numerical approaches were developed to measure the interfacial area between the two fluid phases. Current research in the field of CCS focuses on the design and development of field investigation techniques allowing short response times in on-site plume monitoring and detection. Detection and quantification of the $CO_2$-brine fluid-fluid interface could be realized using Kinetic Interface Sensitive (KIS) tracers as described in Schaffer et al. (2013). Several mathematical models for immiscible two-phase flow and KIS

tracer transport in porous media were developed in Tatomir et al. (2013; 2014), and Tatomir et al. (2015) whereas the latter also considers compositional effects. These modeling approaches require the a priori knowledge of the fluid-fluid-solid system specific capillary pressure $p_c$)-wetting phase saturation ($S_w$)-interfacial area ($a_{wn}$) constitutive relationship (Tatomir et al., 2013). The $p_c$-$S_w$-$a_{wn}$ is an extension to the standard Brooks-Corey model (Brooks and Corey, 1964).

Based on an extended form of Darcy's law (considering fluid-fluid friction force and interfacial forces), the mathematical fundamentals of pore scale two-phase flow regarding the $p_c$-$S_w$-$a_{wn}$ relationship, are given in Hassanizadeh and Gray (1980; 1990; 1993) and Niessner and Hassanizadeh (2008). The derivation of this porous media specific $p_c$-$S_w$-$a_{wn}$ relationship can be realized using physical experiments in form of e.g. micromodels as described amongst others in Karadimitiou et al. (2013; 2014), Karadimitriou and Hassanizadeh (2012) and

Lenormand and Touboul (1988). Corresponding numerical models are given in among others in Ahrenholz et al. (2011) and Porter et al. (2009). These specific approaches are based on Lattice-Boltzmann simulations which are considered as elegant but limited by computational resources (White et al., 2006). Generally, the derivation of this parametric relationship on the pore scale requires a high computational effort as well as a complex numerical model.

This article presents results of a Finite Element Method (FEM) -based pore-scale model to determine the dynamic evolution of the $CO_2$-water interface in geometries with gradually increasing level of complexity. Within this case study, we present the workflow that enables the derivation of $p_c$-$S_w$-$a_{wn}$ relationships of sandstone core sample based model domains. We describe a feasible and relatively simple numerical model that enables the dynamic fluid-fluid interface tracking on μ-CT obtained geometries. Initial and boundary conditions as well as spatial and

temporal discretization are derived using a novel approach taking simplified idealized geometries based on REV-scale pore geometry statistics into account.



The resulting $p_c$-$S_w$-$a_{wn}$ relationships contribute to desiging the KIS tracer laboratory experiments described in (Tatomir et al., 2015). Furthermore, they improve the characterization the Heletz sandstone reservoir (Edlmann et al., 2016; Niemi et al., 2016; Tatomir et al., 2016).

## 2 Heletz Georeservoir Geology and Mineralogy

The sandstone sample on which this case study bases originates from Heletz, Israel, a scientifically motivated deep saline $CO_2$ storage pilot site (Niemi et al., 2016). This injection site extents within depths of 1300 to 1500 m over an area of 22 km². The reservoir itself consists of sandstone layers overlaid by an impermeable cap rock consisting of shale and marl. A detailed description of the site geology is given in Tatomir et al. (2016) and Niemi et al. (2016). With regards to the sample mineralogy, a thin section analysis was carried out to characterize the solid
matrix, which was found to consist of Quartz grains interconnected by a Calcite and Clay cementation matrix. X-ray diffraction confirmed these results as described in Tatomir et al. (2016). Sorting and rounding after Tucker (2011) classified the sample to be very well sorted and rounded with low to high sphericity respectively. The pore space can analogously be defined as homogenously distributed creating a network of pores of mainly similar sizes. Further information gives the later described pore size distribution analysis. A photography of the sample as well
as a thin section and scanning electron microscopy (SEM) images are given in Figure 1.



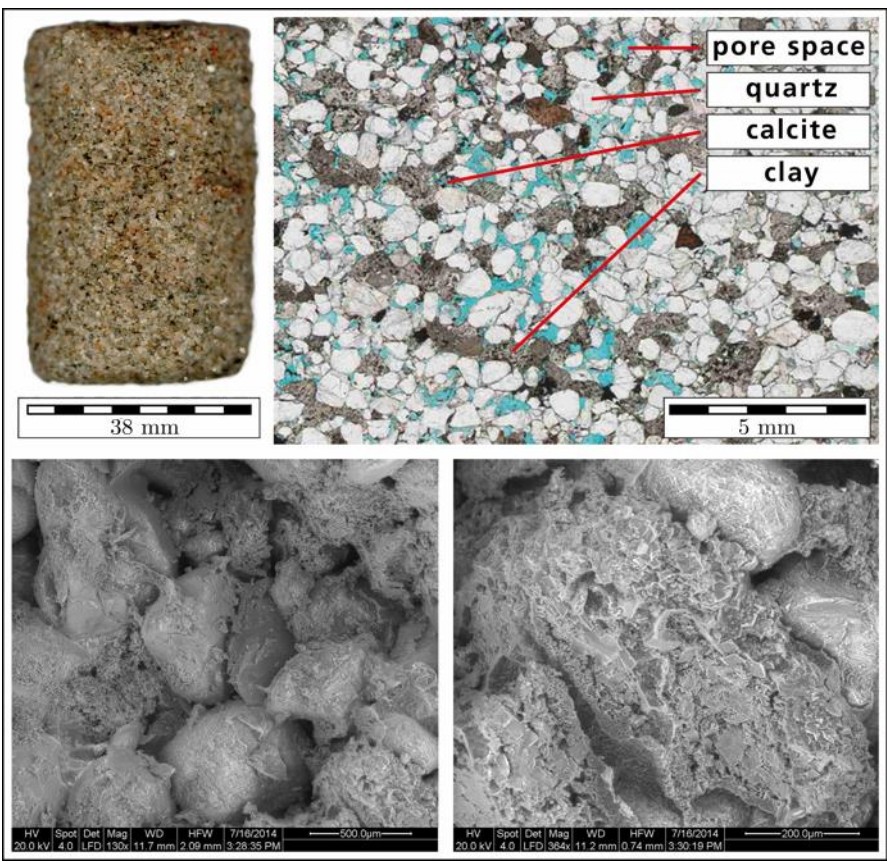

Figure 1: **Top left: Photography of a well core sample. Top right: Thin section image of the sample confirming the well sorted and rounded character. Labels mark the main mineral components quartz, calcite, clay and the pore space. Bottom left: SEM image showing a plane view on a mineral conglomerate of quartz grains interconnected by clayey**

5 **and carbonatic cement. Bottom right: SEM image highlighting the partially significant amount of cementation (modified after Tatomir et al., 2016).**

### 3 Mathematical Model

All modeling in this study was carried out using COMSOL Multiphysics (Version 4.3b). Fluid flow was solved numerically using the incompressible Navier-Stokes Equations (NSE). Defining a Newtonian fluid of constant

10 density and laminar flow enables solving the NSE in form of momentum balance (1) and continuity equation (2) as follows (COMSOL, 2014)

$$\rho \frac{\delta u}{\delta t} + \rho(u \cdot \nabla)u = \nabla \cdot [-pI + \mu(\nabla u + (\nabla u)^T)] + F_{st} + \rho g, \quad \text{with} \tag{1}$$

$$\nabla \cdot u = 0, \tag{2}$$



where $\nabla$ is the nabla operator and t, p, $\rho$, $\mu$, u, g, $I$, T and $F_{st}$ denote time, pressure, density, dynamic viscosity, velocity, gravity, identity matrix, transpose superscript and a surface tension force component respectively. The latter was added in order to include pore scale dependent capillary effects. For simplification, gravity was neglected.

Sharp fluid-fluid interface tracking was realized by implementing the Phase Field Method (PFM). Here, interfaces are defined as regions of rapid change in an auxiliary parameter $P$, or phase field (Popatenko, 2007). Phase change occurs at $-1 < P < 1$.

The change of phase component concentration over time can be described as the product of diffusion coefficient and chemical potential known as Cahn-Hilliard equation (Cahn and Hilliard, 1958). In the present model, this

equation is implemented according to equations 3 and 4 using the terms mobility $\gamma$, predefined interface thickness $\varepsilon$, mixing energy density $\lambda$ and a phase field help variable $\Psi$ (COMSOL, 2014).

$$\frac{\delta P}{\delta t} + u \cdot \nabla P = \nabla \cdot \frac{\gamma \lambda}{\varepsilon^2} \cdot \nabla \Psi \qquad (3)$$

$$\Psi = -\nabla \varepsilon^2 \nabla P + (P^2 - 1)P \qquad (4)$$

The typical interface thickness parameter is $\varepsilon = h_c/2$, whereas $h_c$ is the characteristic spatial mesh element size

at interface position. Calculating the surface tension coefficient $\sigma = \frac{2\sqrt{2}}{3} \frac{\lambda}{\varepsilon}$ in form of energy per area, respectively force per length (Woodruff, 1973) enables to derive the mixing energy density $\lambda = \frac{3\sigma\varepsilon}{2\sqrt{2}}$. The mobility is solved mathematically using $\gamma = \chi \varepsilon^2$ with the manually chosen mobility tuning parameter $\chi$ (here: $\chi = 1$). Volume fractions of fluid 1 (f1) and fluid 2 (f2) can then be calculated according to $V_{f1} = \frac{1-P}{2}$ and $V_{f2} = \frac{1+P}{2}$. Prior to computation, $\rho$ and $\mu$ of both fluids have to be defined. In the phase transition zone, respective $\rho_{mix}$ and $\mu_{mix}$ is

calculated according to

$$\rho_{mix} = \rho_{f1} + (\rho_{f2} - \rho_{f1}) \cdot V_{f2}, \qquad (5)$$

$$\mu_{mix} = \mu_{f1} + (\mu_{f2} - \mu_{f1}) \cdot V_{f2}. \qquad (6)$$





**4 Model Validation**

The NSE model is validated simulating Plane Poiseuille Flow (PPF). PPF describes pressure driven steady, incompressible laminar viscous flow through a 2D channel (Rogers, 1992). Within the channel, a parabola shaped velocity field forms and maximum velocity ($u_{max}$) is found to be in the channel center. This can be calculated using

the following analytical solution derived from the NSE:

$$u_{max} = \frac{h^2}{8\mu}\left(-\frac{\delta p}{\delta x}\right). \tag{7}$$

Here, $h$, $\mu$ and $\frac{\delta p}{\delta x}$ denote the channel height, dynamic viscosity and change of pressure over channel length respectively. More detailed descriptions about PPF are given among others in Chughtai et al. (2009) and Shahriari et al. (2013). Comparing results of the analytical solution with the corresponding $u_{max}$ obtained from modeling

(based on a channel geometry of length and height of approx. $x = 146\ \mu m$ and $h = 48\ \mu m$, whereas the latter represents the mean pore diameter of the complex μ-CT based geometry) gave a deviation of $\Delta u_{max} = 0.05\%$. Thus, the computation of PPF can be seen as precise. Here has to be stated, that the channel length was extended on both sides by a correction length $x_{corr} = 1/2 \cdot x$ in order to neglect boundary in- and outflow effects. These extensions were neglected for the validation analysis. A visualization of the model setup as well as model results

in form of isolines of velocities is given in Figure 2.

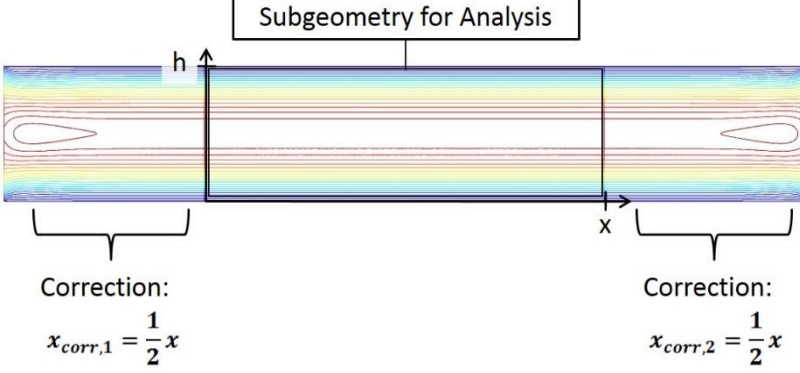

**Figure 2. Model setup for the model validation using Plane Poiseuille Flow. Colored lines indicate isolines of velocity, whereas blue and red colors define lower and higher velocities respectively. The subgeometry used for the analysis is marked by the black box.**

For the validation of the PFM regarding the emulation of the sharp fluid-fluid interface, Rayleigh-Taylor Instability (RTI) has been numerically designed and modelled. In principle, RTI is described as the gravity driven unstable displacement of a light fluid $f_2$ with a heavy fluid $f_1$ (Lord Rayleigh and Strutt, 1883; Taylor, 1950; Mikaelian, 2014). Initial separation of both fluids is given by a cos-shaped interface of the amplitude $\eta_0$. Over the course of



time, $f_1$ breaks into $f_2$ causing a characteristic interface amplitude change $\delta\eta/\delta t$. With regards to this characteristic $\delta\eta/\delta t$, Mikaelian (2014) defines three stages of RTI in form of linear, nonlinear and turbulent regime. At transition of nonlinear to turbulent regime (later time steps at $\eta(wavelength\ \lambda) = \frac{\lambda}{2\pi}$, Anuchina et Al., 1978; Kordilla, 2014), the interface amplitude change can be described mathematically as follows (Read, 1984;

Youngs, 1984):

$$\eta(t) = \alpha_q A g t^2. \tag{8}$$

$A$, $g$ and $t$ are Atwood number (dimensionless density ratio: $A = {[\rho_1 - \rho_2]}\big/{[\rho_1 + \rho_2]}$), gravitational acceleration and time respectively, while $\alpha_q$ is a growth rate coefficient that should range within 0.01 and 0.08 (Glimm et al., 2001). Modelling was carried out within a geometry based on the auxiliary parameter $d_p = 0.048\ m$ describing a

rectangular shape of a width $w = d_p$ and height $h = 2d_p$. The initial cos-shaped interface was defined as a function $y = d_p + 0.05 * d_p * \cos(x)$ with a vector $x = (0, \frac{\pi}{4}, \frac{\pi}{2}, \frac{3\pi}{4}, \pi, \frac{3\pi}{4}, \frac{\pi}{2}, \frac{\pi}{4}, 0)$ separating $f_1$ representing $H_2O$ ($\rho_{H2O} = 999.6\ kg/m^3$) and $f_2$ as supercritical $CO_2$ ($\rho_{CO2} = 700\ kg/m^3$). Boundary conditions were set to no slip and modelling was solved transiently over a time interval $t = 0.75\ s$ using a direct MUMPS (MUltifrontal Massively Parallel sparse direct Solver) solver. Schematic model setup and model outcome is visualized in Figure

3.

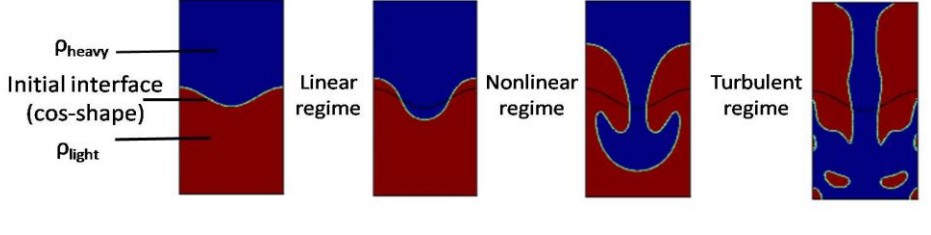

**Figure 3.** **Schematic model setup and phase propagation over time of the model validation using Rayleigh-Taylor instabilities. All RTI stages in form of linear, nonlinear and turbulent regime are visualized. Blue and red colours represent heavier ($\rho_{heavy}$ here: $H_2O$) and lighter fluid ($\rho_{light}$ here: $CO_2$) respectively. For comparability, the cos-shaped**

**initial interface is given continuously as a black line, while the propagating interface is illustrated in grey colour.**

Regarding the results, a $\alpha_q = 0.068$ was found to be a good fit. Comparing the modelling result with the corresponding analytical solution gives a $R^2 = 0.99$. The associated plot is given in Figure 4. In conclusion, the model agrees to the analytical solution to a very high degree.



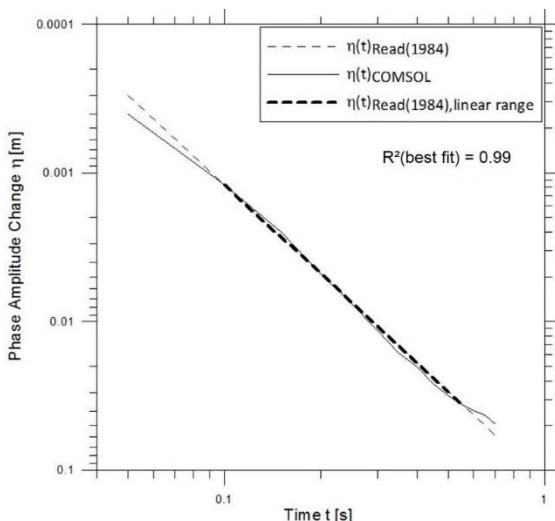

**Figure 4:** Interface amplitude change $\eta$ of Rayleigh-Taylor instability over time. The $\eta_{COMSOL}$ represents the COMSOL modeling result as a grey line while $\eta_{Read}$ represents the analytical solution for the late nonlinear to early turbulent RT regime as a dashed line. The bold dashed line visualizes the representative part of the analytical solution.

Further model validation with regards to NSE and PFM in form of 2D capillary rise within a tube was carried out. Here, fluid rises within a capillary contracting air due to imbalanced intermolecular adhesive forces between liquid and solid and intermolecular cohesive forces within the liquid. The corresponding analytical solution derived for the fluid column height $h$ is given in the Young-Laplace equation (Young, 1805; Laplace, 1805; Norde, 2003) as follows:

$$h = \frac{2\gamma\cos(\theta)}{\rho g r}.$$   (9)

$\gamma, \theta, \rho, g$ and $r$ denote surface tension, contact angle, fluid density, gravitational acceleration and capillary radius respectively. Modelling was carried out on a geometry consisting of a rectangular base of height and width of $2 \cdot 10^{-3}$ m respectively $4 \cdot 10^{-3}$ m and an overlaying capillary extending a height of $40 \cdot 10^{-3}$ m and a width of $2 \cdot 10^{-3}$ m. Pressure driven replacement of air of $\rho_{air} = 1.2 \ kg/m^3$ by $H_2O$ ($\rho_{H2O}$ as above) was simulated, whereas

a capillary pressure of $p_c = 145 \ Pa$ was defined on one lateral side of the rectangular base (inlet) below the capillary and a pressure of zero at the capillary top. Remaining boundaries were set to no slip. The model was solved transiently over a total time of 2 s using the iterative Generalized Minimal Residual Method (GMRES). Model setup and capillary rise at pressure equilibrium ($t = 2$ s) are illustrated in Figure 5.





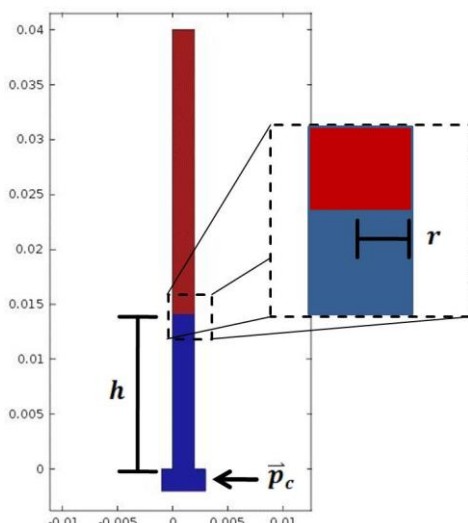

**Figure 5:** **Model setup for the model validation via Young-Laplace capillary rise at pressure equilibrium. Blue and red colours define water and air respectively. Height of water column is $h$ at a capillary radius $r$ and an inlet capillary pressure $p_c$. Orientation axes values are given in m.**

Comparing the model results in form of height of water column in the capillary with the corresponding solution of equation 9 gave a deviation of $\Delta h = 4.67$ %, which can be seen as sufficiently accurate. The mathematical model used within this study can be seen as successfully validated.

**5 Numerical Workflow**

    The numerical workflow steps (e.g., generation of the μ-CT based DSA, statistical analysis of the pore space
properties, the two-phase flow model setup, etc.) are described in the following section.

In order to test and modify the FEM-based model described in section 3 on the sample specific pore geometries (e.g.**,** transitions from big pores to small bottlenecks), statistical parameters of the μ-CT-based digital sample approximation (DSA) had to be determined. **A** porosity based REV-analysis was carried out in order to derive the domain size for a pore size distribution (PSD) analysis. Subsequent implementation of a PSD-analysis gave
statistical information about average-, maximum- and minimum pore diameters. This statistical information was used for designing idealized model domains and for extracting subgeometries from the DSA. As a next step, the model was implemented stepwise, first on simple geometries (idealized single pore, idealized pore network) and later on complex geometries (μ-CT-based pore, μ-CT-based pore network). This enabled to test the model performance with regards to very steep pressure gradients (and critical flow) building up on small bottlenecks. If
the equation system could not be solved for a given geometry, modifications in the model were done. These



modifications include choice of temporal discretization, initial- and boundary conditions and in extreme cases simplifications in the mathematical model (e.g., neglecting the contact angle by changing apparent slip to no slip boundary conditions). This was done until a model and model setup for the successful implementation on complex µ-CT-based geometries was derived. Figure 6 illustrates the workflow schematically.

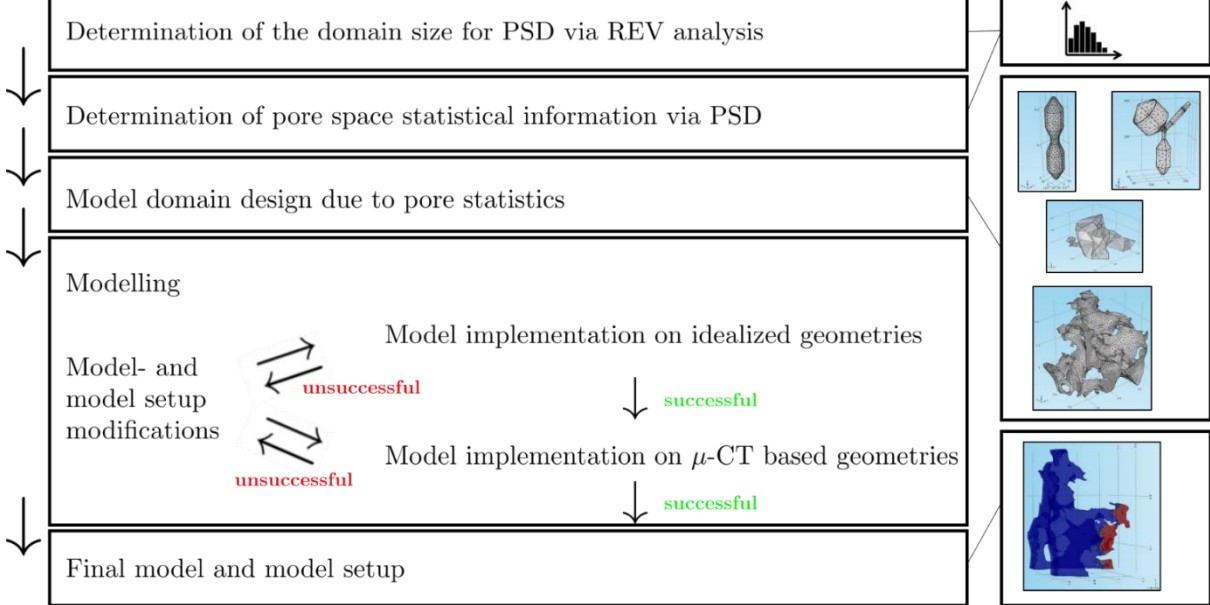

Figure 6: **Schematic overview of the workflow for deriving the final two-phase flow model and model setup for complex µ-CT-based model domains.**

**6 Generation of the µ-CT-based DSA, Geostatistical Analyses, Model Domain Description and Initial Model Setup**

10    **6.1 Domain Construction and Mesh Generation via µ-CT Imaging**

The 3D imaging of the sandstone has been performed with a nanotom S 180 µ-CT (180 kV, 500 mA) device, located at the Leibniz Institute for Applied Geophysics (Department 5 -Petrophysics and Borehole Geophysics) in Hannover. The nanotom is a compact CT system for pore scale imaging purposes, i.e. for high resolution imaging within the micrometer (typically 1-2 µm) to sub-micrometer range (about 700 nm for very small samples),

15    featuring high image sharpness due to a significantly reduced penumbra effect (Brunke et al, 2008). The image data was processed with the AVIZO Fire software suite ([www.fei.com/software/avizo3d/](www.fei.com/software/avizo3d/)), including scan artifact reduction and image filtering. Due to the low image noise and due to the fact that only single phase segmentation of the pore system was needed, segmentation was performed by the fast and robust "automatic threshold selection

method" as described by Otsu (1979). Afterwards, the pore space was transferred into a mixed hexagonal-tetrahedral grid by using the extended volume marching cubes algorithm as implemented within the software ScanIP and as described by Young (2008). This step included an optimization of the segmented grid, which significantly reduced the number of grid cells (almost by a factor of 10) and hence computational time and power,

without "destroying" topological information of the grid-surface. The final mesh has been exported as an ASCII-grid (*.mphtxt), which has been imported to COMSOL for modeling.

## 6.2 REV-Analysis

The interfacial area between the two fluid phases for the macro-scale models is represented as volume averaged. Therefore, the representative elementary volume (REV) with regards to effective interconnected porosity was

identified and extracted from digital sample approximations. Further information about the REV can be found in Bear (1988). A detailed description of the workflow and results specifically for Heletz sandstone cores is given in Tatomir et al. (2016).

Results indicate a REV-range of a subvolume edge length band width ranging from approx. 120 voxel to 280 voxel. Within this range, fluctuations of porosity are minimal. Subvolumes bigger than 280 voxel edge length

show an increase in porosity and can be identified as domain where macroscopic effects are dominant. Subvolumes smaller than 120 voxel can be further distinguished in a transition phase ranging from 50 to 120 voxel where fluctuations in porosity are relatively low and in domains smaller 50 voxel domain where microscopic effects are dominant and fluctuations are high.

## 6.3 Pore Size Distribution Analysis

Based on the μ-CT images, a series of operations combining pore identification, binarization, separation and statistical analysis regarding geometric extent of features was implemented on an arbitrarily extracted REV-scale subvolume of the DSA using the software Avizo Fire (Version 8.1). Pore identification and characterization was based on their specific greyscale value. Further separation was done according to the watershed transformation technique as described amongst others in Sheppard et al. (2005) and Soille (2003). Finally, all relevant data was

determined by applying the special 'label analysis' operation. The results of the PSD are given in form of a histogram (Figure 7) and Table 3 in the appendix, whereas the latter summarizes most important statistical information, such as minimum-, maximum- and mean pore diameters. A detailed description of the workflow is given in Tatomir et al. (2016).



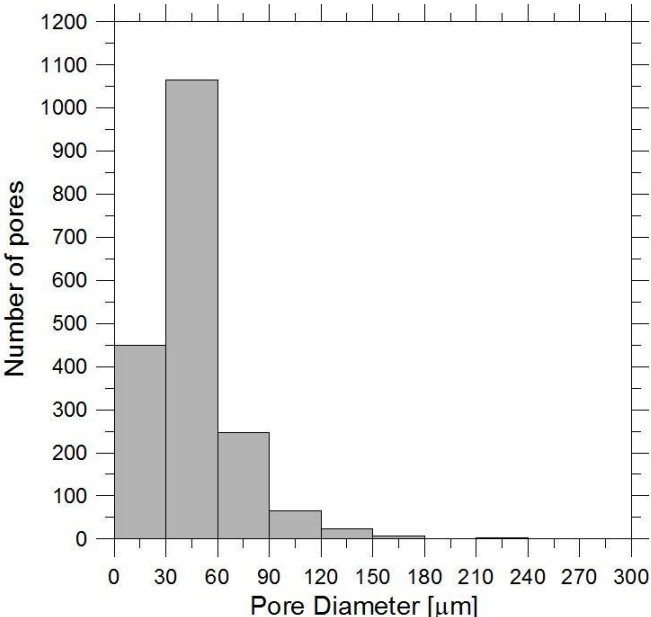

**Figure 7: Result of the pore size distribution for an REV-scale subvolume arbitrarily placed in the sample approximation as number of pores vs. pore diameter, whereas pore diameters were unified in 30 µm intervals. Due to representation, this figure excludes the pore representing the maximum pore diameter $d_{Pore,max}$, which exceeds the pore diameter range significantly.**

It can be observed that the majority of pores are represented by diameters up to 90 µm. In fact, approximately 95% of pores are below a threshold of the doubled mean pore diameter. This value was considered instead of the maximum pore diameter when designing the idealized geometries due to optimized geometry meshing. It has to be stated that remaining bigger pores may be few, however these pores play a more significant role for fluid flow representing preferential flow paths which might conduct increased fluid volumes at smaller travelling times (Lal et al., 2004). Beside the doubled mean pore diameter $2\bar{d}_{Pore}$, idealized model domain design as well as subdomain extraction from the DSA are based on $\bar{d}_{Pore}$ and $d_{Pore,min}$.

### 6.4 Model Domains and Initial Model Setup

Model domains are called idealized pore, idealized pore network, µ-CT-based pore and µ-CT-based pore network. A detailed description is of each domain is given in the following:

The idealized pore measures the mean pore diameter $(\bar{d}_{pore})$. A geometric bottleneck with a $d_{throat} = \frac{1}{2} \cdot \bar{d}_{pore}$ was positioned in the middle of the domain in order to test pore throat in- and outflow effects. Domain in- and outlet were scaled to $d_{in,out} = \frac{1}{4} \cdot \bar{d}_{pore}$.



The idealized pore network domain led from a pore with a diameter $\bar{d}_{pore}$ simultaneously into two pores, each represented by the minimum pore diameter ($d_{min}$) and the doubled mean pore diameter ($2 \cdot \bar{d}_{pore}$). The latter is approximated as a maximum pore diameter for the 95th percentile of pore diameters in the DSA. Pressure gradient driven flow was conducted z-axially for both idealized geometries. Flow in and outlet were positioned at $z_{in} =$

$0 \, \mu m$ and $z_{out} = z_{max}$ respectively and the initial interface was located at $z = 50 \, \mu m$ and $z = 10 \, \mu m$ for the idealized pore and the idealized pore network respectively.

The µ-CT-based pore geometry was explicitly chosen due to its pore diameter which approximates $\bar{d}_{pore}$ with $d \approx$ $40 \, \mu m$. Furthermore, it was chosen due to the presence of a dead end pore in order to observe trapping effects. The µ-CT-based pore network geometry was chosen due to its band width of pore diameters $25 \, \mu m \lesssim d \lesssim$

$220 \, \mu m$ approximating the corresponding band width of the REV-scale µ-CT geometry. Again, the presence of dead end pores was considered. Pressure driven flow was conducted y- and z-axially for the µ-CT-based pore geometry and the µ-CT-based pore network geometry respectively. Domain in and outlet were set analogously to the idealized geometries and the initial interfaces were positioned at $z = 0.1 * z_{max}$ and $z = 0.25 * z_{max}$ respectively. Regarding the domain pore sizes, both µ-CT-based geometries directly relate to geometric and later

flow characteristics that are present in the REV-scale geometry. All geometries at initial condition are shown in Figure 8.





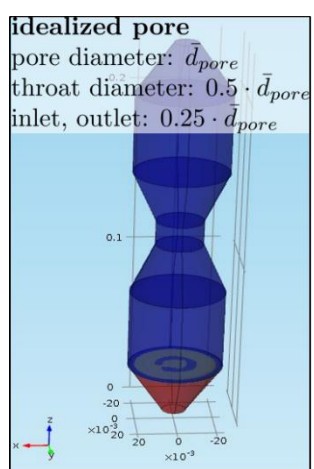
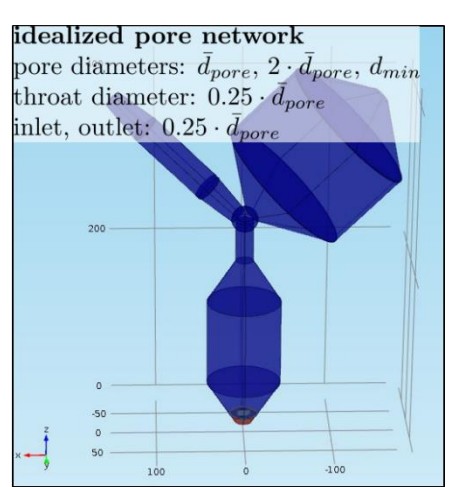
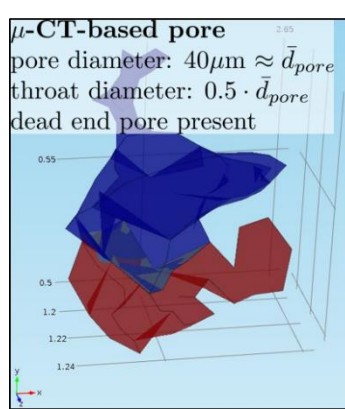
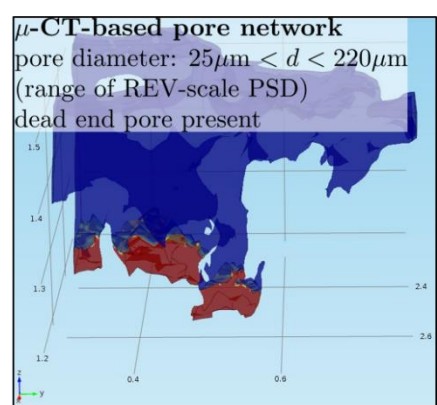

**Figure 8:** **Heletz sandstone DSA based idealized and µ-CT-based model domains used within this study. All domains are visualized at initial condition. Blue colors represent water, red colors $CO_2$. The orientation grid is given in µm-scale for the idealized pore network domain and for all other domains in mm.**

5    The fluid properties for supercritical CO2 and water are given in Table 1.

**Table 1. Fluid properties representing supercritical $CO_2$ and $H_2O$ in the models.**

|  | $H_2O$ | $CO_2$ |
|---|---|---|
| Density $\rho$ [kg/m³] | 973.9 | 470 |
| Dynamic viscosity $\mu$ [Pa s] | 3.59e-4 | 0.55e-4 |

The temperature is set to 333.15 K which is the measured temperature in Heletz reservoir (Niemi et al., 2016). Analogous to typical on-site injection, a one-axial pressure drop was chosen to induce fluid flow, whereas the overall pressure gradient was defined as capillary pressure ($p_{inlet} - p_{outlet} = p_c$). Domain in- and outlet were





defined as constant pressure head (Dirichlet) boundary condition ($p = const.$) and viscous stress was neglected [$\mu(\nabla u + (\nabla u)^T)$=0]. The model was implemented on a range of capillary pressures $1\,MPa < p_c < 15\,MPa$ with a $\Delta p = 2\,MPa$, whereas the maximum $p_c$ approximates a field injection shut-in pressure. In order to take wetting and nonwetting-fluid effects into account, a contact angle $\theta = 30° = \pi/6\,rad$ was introduced. This value for a

$CO_2$ (supercrit.)-$H_2O$-$SiO_2$ system at a $p_c = 10$ MPa originates from a study conducted by Espinoza and Santamarina (2010). Here has to be stated that effects due to a change in $\theta$ are of importance when quantifying $a_{wn}$ on a pore scale. A change in contact angle results in a change in interfacial curvature (Jiang and Zeng, 2013). An increase in $\theta$ leads to a bigger curvature, which leads to an increase in $a_{wn}$ and vice versa. The computation of $\theta$ could be realized by defining an apparent slip boundary condition on all remaining boundaries. This enforces a

zero velocity at the wall. The product of velocity u and interface normal (using the previously introduced auxiliary phase field parameter) $n = \frac{\nabla P}{|\nabla P|}$ is zero.

$$u \cdot n_{wall} = 0 \tag{10}$$

Such an apparent slip introduces a larger velocity gradient $\Delta u$ at the boundary while the fluid speed seems (but in fact does not) to extrapolate to zero below the surface (Granick et al., 2004; Forster and Smith, 2010). In this

model, an apparent slip is enabled when adding the boundary condition 'wetted wall'. This boundary condition adds a frictional boundary force $F_{fr}$

$$F_{fr} = -\frac{\mu}{\beta}u, \tag{11}$$

whereas $\beta$ is slip length and $\mu$ viscosity. This enables to add a boundary integral (COMSOL, 2014)

$$\int_{\delta\Omega} test(u) \cdot [\sigma(n_{wall} - (n \cdot \cos(\theta))\delta)]dS, \tag{12}$$

which consists of the surface tension coefficient σ and the delta function $\delta = 6|P(1 - P)||\nabla P|$. The test function $test(u)$ enables to neglect (12) when applying a no slip boundary condition. Subsequently, this setup enables defining $\theta$.

### 6.5 Numerical Model Setup (after Model and Model Setup Modifications)

For the model based on idealized geometries, an appropriate time step size was found to be $\Delta t = 5 \cdot 10^{-8}\,s$. Total

time intervals were set to $t_{max} = 1 \cdot 10^{-4}\,s$ and $t_{max} = 1 \cdot 10^{-5}\,s$ for the idealized single pore and idealized pore network respectively.





Implementing the initial model and -model setup on the μ-CT-based geometries was only possible after modifications. The wetted wall boundary condition had to be changed to a no slip boundary condition in order to avoid imbibition effects. The contact angle was neglected and thus not considered for later results. The total time interval had to be decreased when applying bigger capillary pressures. In case of the pore benchmark geometry,

the initial time interval of $t_{max} = 5 \cdot 10^{-5}\ s$ was reduced to $t_{max} = 1 \cdot 10^{-5}\ s$ for each $p_c \geq 5 \cdot 10^6\ Pa$. In case of the pore network benchmark geometry, the time interval choice was more variable. It is given in Table 4. Analogous to above, time step size was kept at $\Delta t = 5 \cdot 10^{-8}\ s$. In order to minimize computation time, the absolute error tolerance of the solver was changed from 0.001 to 0.01. While the μ-CT-based pore geometry modelling could be successfully executed on these settings, model performance on the μ-CT-based pore network

geometry had to be stabilized by introducing a Dirichlet-type constant outlet pressure boundary condition. This setting enabled to solve the equation system and avoid otherwise appearing singularities. Choice of this pressure was found to depend on the inlet capillary pressure. Complete displacement of $H_2O$ was not possible in the pore network benchmark, which is due to trapping effects. Resulting minimum wetting phase saturations $S_{w,min}$ for each $p_c$ are given in Table 2.

**Table 2. Outlet pressure $p_{out}$, maximum model time interval $t_{max}$ and minimum reached wetting phase saturation $S_{w,min}$ for each corresponding capillary pressure $p_c$ for the pore network benchmark model.**

| $p_c$ [Pa] | $p_{out}$ [Pa] | $t_{max}$ [s] | $S_{w,min}$ [-] |
|---|---|---|---|
| 0.96e6 | 4e4 | 2.1e-4 | 0.27 |
| 2.96e6 | 4.5e4 | 0.46e-4 | 0.45 |
| 4.95e6 | 5e4 | 1e-4 | 0.22 |
| 6.75e | 2.5e5 | 0.64e-4 | 0.29 |
| 8.85e6 | 1.5e5 | 0.43e-4 | 0.37 |
| 10.7e6 | 3e5 | 0.79e-4 | 0.17 |
| 12.7e6 | 3e5 | 0.56e-4 | 0.27 |
| 14.7e6 | 3e5 | 0.8e-4 | 0.17 |

### 6.6 Mathematical Derivation of the $p_c$-$S_w$-$a_{wn}$ Relationship

Since the pressure drop within the domain was kept constant for each simulation, the calculation of $p_c$ was realized

by subtraction of outlet from inlet pressure according to



$$p_c = p_{inlet} - p_{outlet}. \tag{13}$$

The wetting phase saturation $S_w$ was determined by dividing the volume integral of wetting phase filled domain cells $B_w$ about the total the total domain volume (volume integration $\iiint_V 1 \, dV$). This is mathematically expressed as follows

$$\quad S_w = \frac{\iiint_V B_w \, dV}{\iiint_V 1 \, dV}. \tag{14}$$

In order to quantify the specific interfacial area, an isosurface data set was generated. This data set represented the 0.5 concentration of each, nonwetting and wetting phase. The corresponding auxiliary phase field parameter $P$ is zero. All cells represented by this isosurface are denoted with $B_{P=0}$. A subsequent surface integration of this isodata set normalized to the total domain volume gave $a_{wn}$ as follows

$$\quad a_{wn} = \frac{\iint_V B_{P=0} \, dV}{\iiint_V 1 \, dV}. \tag{15}$$

## 7 Results

### 7.1 Dynamic evolution of CO2-water interface

An exemplary time series of phase as well as interface propagation within the μ-CT-based pore network geometry is given in Figure 12. Same is given for the μ-CT-based single pore geometry as Figure 11. Complete $H_2O$ replacement ($S_w = 0$) could not be achieved in any of the given domains, which is partly due to trapping and bubble forming effects. A more detailed description about such effects is given in Saeedi (2012). Residual water in dead end pores and bubbles can be observed in the last time step visualized in both Figures 9 and 10.





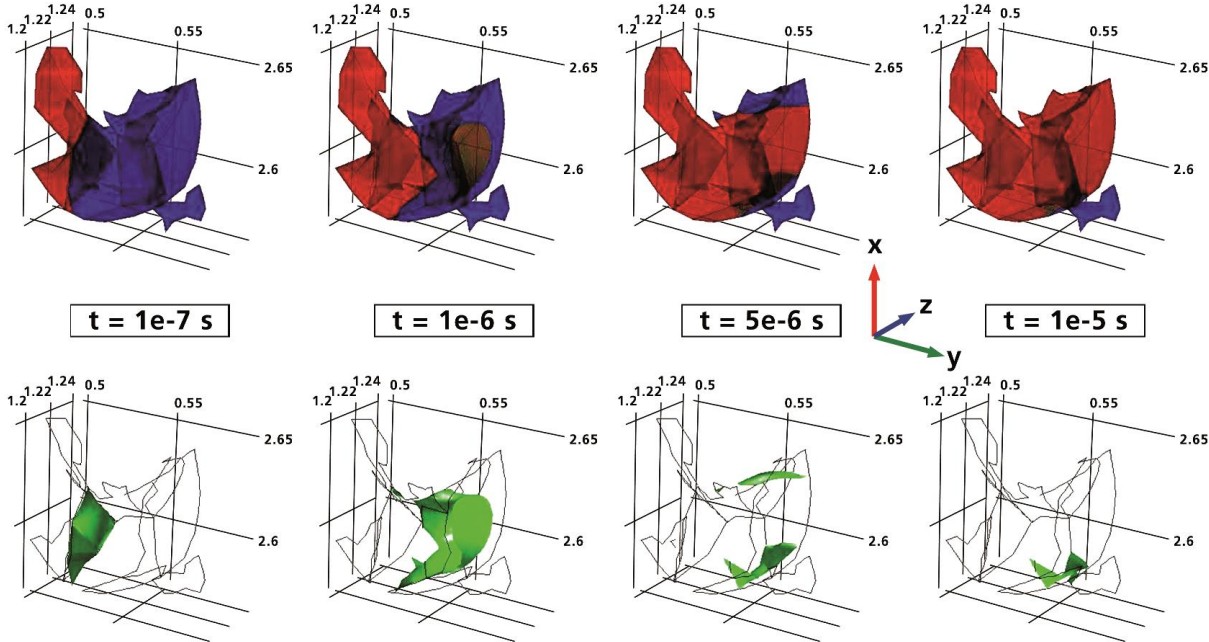

**Figure 9:** Image series showing the propagation of supercritical $CO_2$ displacing $H_2O$ (above) as well as the corresponding propagation of the fluid-fluid interface (below) at $p_c = 15e6\ Pa$. The series is given for selected time steps within the μ-CT-based single pore geometry. Blue, red and green colours represent $H_2O$ and $CO_2$ phase as well as fluid-fluid interface respectively. Grid values are given in mm.




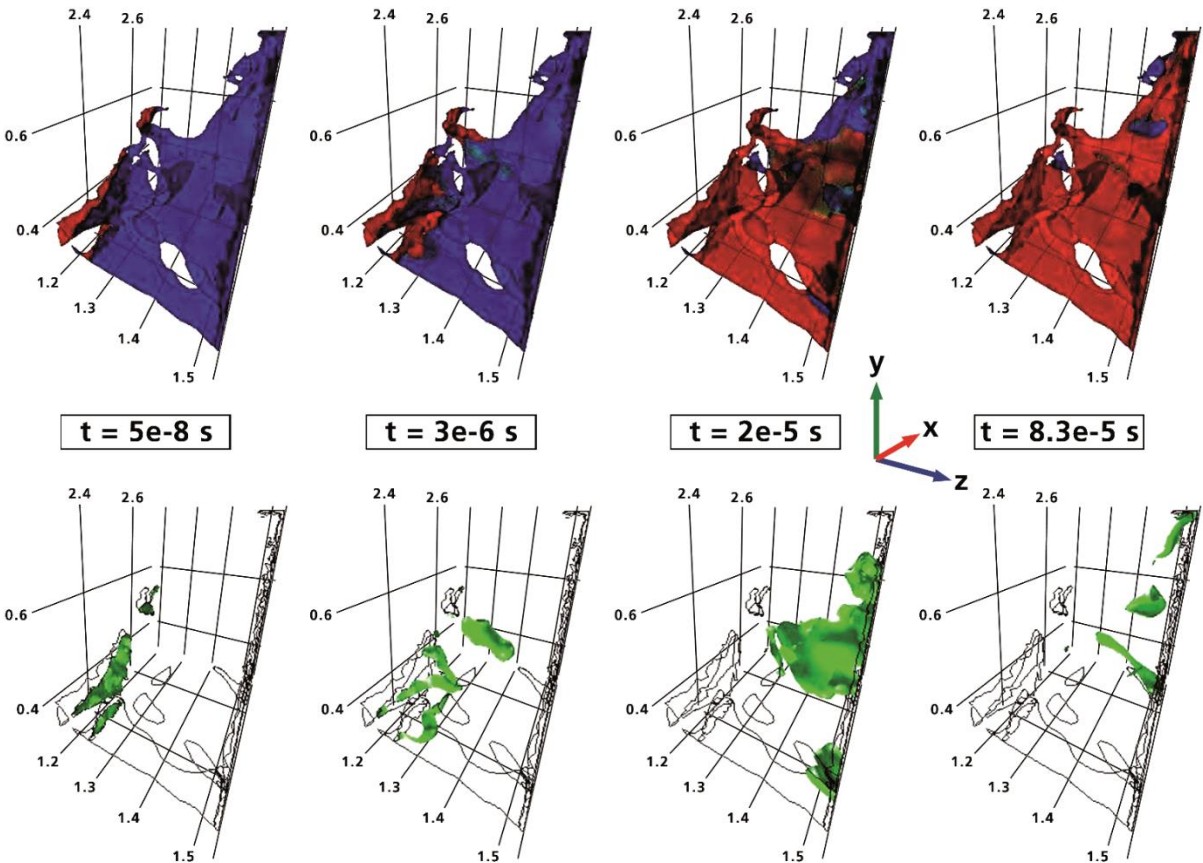

Figure 10: **Image series showing the propagation of supercritical CO₂ displacing H₂O (above) as well as the corresponding propagation of the fluid-fluid interface (below) at** $p_c = 15e6\ Pa$**. The series is given for selected time steps within the μ-CT-based pore network geometry. Blue, red and green colours represent H₂O and CO₂ phase as well as fluid-fluid interface respectively. Grid values are given in mm.**



Figure 11 visualizes the change of $S_w$ over time for μ-CT-based pore network geometry all applied capillary pressures. The figure is given in log-log axis scale and the initial wetting phase saturation for all scenarios ($S_w = 0.93$) at time step zero is neglected.

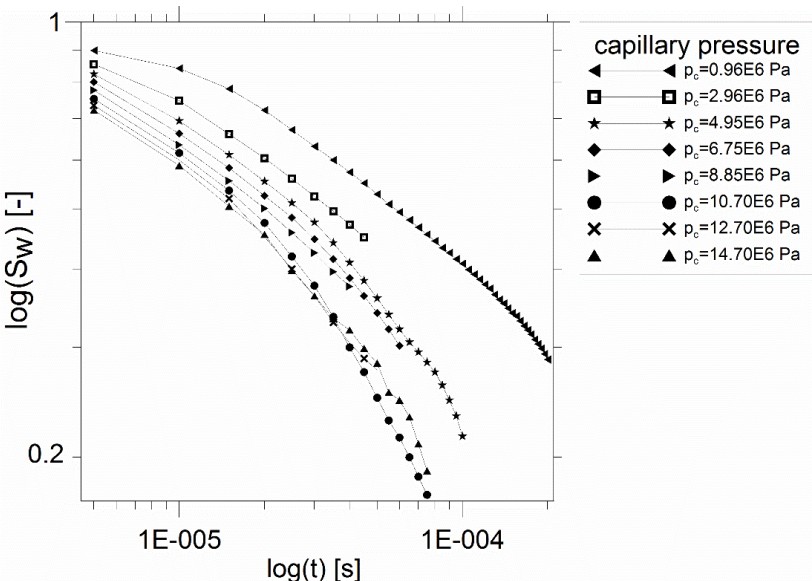

5 Figure 11: **Plot of $S_w$ over $t$ for each $p_c$.**

It can be clearly observed that an increased capillary pressure results in a significant decrease of wetting phase saturation within the domain. However, at pressures $p_c > 10.70\ Pa$, fluctuations in $S_w$ at later time steps are visible, which might be due to the high difference in fluid density of both phases (Joekar-Niasar et al., 2010).

**7.2 $p_c$-$S_w$-$a_{wn}$ relationships for the μ-CT-based geometries**

10 Figure 12 illustrates the $p_c$-$S_w$-$a_{wn}$ relationship obtained for both μ-CT-based geometries. Differences within the curves underline the highly geometry-specific character at different domain scales.





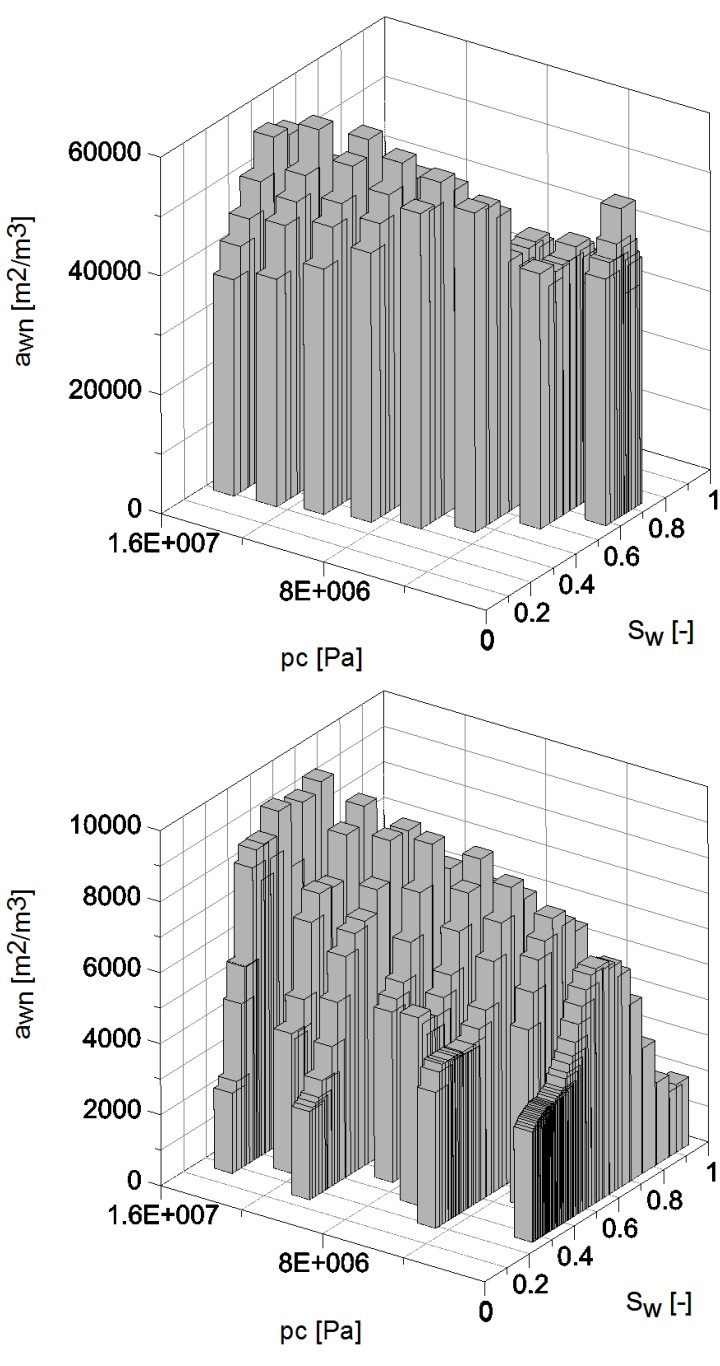

Figure 12: **3D bar plots of the $p_c$-$S_w$-$a_{wn}$ relationship obtained from the µ-CT-based pore geometry (above) and the µ-CT-based pore network geometry (below). Note the different scaling in $a_{wn}$.**



Given $p_c$-$S_w$-$a_{wn}$ relationships show that the maximum observed interfacial area correlates to an increase in capillary pressure. Peaks in $a_{wn}$ distribute for all $p_c$ approximately at a range $0.7 > S_w > 0.35$. It can be stated, that both relationships show the typical bell-shaped form, which is widely postulated in literature (Joekar-Niasar et al. 2010; Karadimitriou and Hassanizadeh, 2012; Karadimitriou et al., 2013).

**8 Discussion**

A case study on Heletz sandstone was carried out to present and test a workflow for determining the constitutive capillary pressure-saturation-interfacial area relationship based on µ-CT obtained geometries. Based on the DSA specific geostatistics, modelling domains were designed and extracted. Modelling on these domains enabled to test the model and model setup on PSA specific steep pressure gradients (at small bottlenecks) causing critical flow and problems solving the equation system. To avoid the latter, the presented workflow can be seen as a systematic procedure to successively modify the implemented two-phase flow NSE model towards a successful model implementation for such a type of modelling problem.

The design of the idealized and of the extracted µ-CT-based geometries can be improved by taking the whole PSD (including pores representing e.g. quantiles) into account and not only average-, maximum and minimum pore sizes. For the modelling problem within this case study, a two immiscible fluid phase system consisting of water and $CO_2$ is considered. For practical field applications, prior to injection, the Heletz aquifer is filled with brine which has a higher density than pure water, and a different viscosity. Thus, the model does not represent field conditions. However, envisioned laboratory experiments (e.g., Tatomir et al, 2015) require the determination of the $p_c$-$S_w$-$a_{wn}$ relationships. Therefore, the main objective of this case study was to develop a feasible workflow for obtaining $p_c$-$S_w$-$a_{wn}$ relationships from µ-CT-based geometries of the Heletz sandstone.

The scanned sandstone sub-core samples do not represent in situ reservoir conditions, such as the pore space compaction occurring at reservoir depth (below 1600 m), which implies reduced rock permeability and porosity. The digital sample approximations used for later modeling represent ex situ rock characteristics. Comparing permeability results from unpressurized and pressurized core samples, Tatomir et al., (2016) shows am approximate overestimation of permeability by a factor of 1.5 (on the same investigated core samples). However, when compared with the permeability results of Hingerl et al., (2016), who used a 9 MPa pressurization, the overestimation reaches a factor of 4. Furthermore, the comparison of capillary pressure – saturation relationships obtained from µ-CT modeling and from laboratory results mercury porosimetry showed important differences (Tatomir et al., 2016).



Furthermore, the accuracy of mesh generation via µ-CT scan must be seen as a function of scanning resolution. Micro-scale features below scanning resolution are lost and subsequent effects of these features on flow characteristics cannot be taken into account. From the mineralogical investigations of the Heletz sandstone the presence of a clay coating on quartz grains was observed. Dry clay minerals are known to swell when in contact

with a fluid like water. This process undergoes with a further change in pore space characteristics, which was neglected for simplification.

The composed numerical model could be successfully validated. NSE was simplified by defining flow incompressibility. Pressure driven flow at steep pressure gradients on in highly irregular geometries lead to the formation of an extremely heterogeneous flow velocity field. Narrow geometric features such as bottlenecks result

in extremely steep pressure gradients and high flow velocities. Velocities were often found to exceed the fluid specific speed of sound $c$. Calculations based on supercritical $CO_2$ in fully $CO_2$ saturated locations showed a maximum found exceedance of $v \approx 8 * c$ ($MACH \approx 8$). Since incompressibility can be assumed when $v < c$ (Wesseling, 1991), this threshold is considerably exceeded and critical flow must be implied. Hence, the neglecting of compressibility leads to model inaccuracies. Same applies for the model simplification defining flow to be

laminar. At narrow geometric features, calculated Reynolds numbers exceed the threshold for turbulent flow by an approximate factor of 6.7 ($Re_{max} \approx 20190$). Thus, neglecting turbulent flow leads to further model inaccuracies. A further source for inaccuracies can be seen in defining fluid-fluid immiscibility. As described by Tassaing et al. (2004), conditions of water dissolved in $CO_2$ under modeling conditions would range up to several mol per liter stating miscibility. Modifications on the mathematical model and model setup carried out during the

workflow, such as the negligence of the contact angle were done in order to be able to solve the equation system. The introduction of a judiciously chosen background pressure enabled to avoid extremely steep pressure gradients at narrow geometric features. At this point, modifications and simplifications must be seen as inevitable under the given circumstances. Simplifications including the negligence of compressibility, miscibility and turbulent flow were done in order to avoid a considerable increase in model complexity. Avoiding these simplifications would

have led excessive computational effort and long computation times. Regarding this and the fact that this study aimed at presenting a feasible, easily applicable tool for the tracking and quantification of the dynamic fluid-fluid interface for a very challenging modelling problem with high (geometric) complexity, such simplifications must be seen as a necessity.



## 9  Conclusions

In this study we have developed a numerical workflow for determination of capillary pressure-saturation-fluid-fluid interfacial area on highly complex μ-CT obtained geometries. The presented workflow consists of a successive number of model runs on geometries with increasing degree of complexity.

The $p_c$-$S_w$-$a_{wn}$ relationship is obtained from modelling idealized geometries with average, maximum and minimum pore size and estimated pore throats resulted from PSD. Pore-throats are important as they control the entrance of the non-wetting phase in the fully wetting phase saturated domain. This is defined at macro-scale (REV-scale) as the entry pressure.

We have not conducted any comparison of with the real REV geometry because the geometry was too complex to

resolve with the two phase NSE model. However, by preserving the average properties of the medium and the average pore-throat diameters it is attempted to obtain the same kind of accuracy. Future work will concentrate in simulating larger domains on bigger clusters and determine the accuracy of the technique. Also, no comparison with a pore-network models was conducted, but it is known that the idealizations imposed by such the pore-network models also results in incorrect estimation of the pc-S (e.g., Tatomir et al. 2016, Leu et al., 2014,

Wildenschild et al., 2005, Wildenschild and Sheppard, 2013).

Last but not least, these results contribute to improving the characterization of the Heletz reservoir where $CO_2$ injection operations are planned within the next year.

### Acknowledgements

We want to show our gratitude to the three reviewers for improving the quality of this manuscript. We want to

thank Juliane Herrmann (LIAG) for figure composition and visualization, Stephan Kaufhold (Federal Institute for Geosciences and Natural Resources, BGR) for the SEM analysis and Insa Neuweiler for her helpful insights.

### List of Abbreviations

|       |                                                   |
|-------|---------------------------------------------------|
| CCS   | Carbon Capture and Storage                        |
| DSA   | Digital Sample Approximation                      |
| FEM   | Finite Element Method                             |
| GMRES | Generalized Minimal Residual Method               |
| KIS   | Kinetic Interface Sensitive (Tracers)             |
| MUMPS | Multifrontal Massively Parallel Sparse Direct Solver |
| μ-CT  | Micro-Computed Tomography                          |



| NSE | Navier-Stokes Equations |
|---|---|
| PFM | Phase Field Method |
| PMD | Porous Medium Domain |
| PPF | Plane Poiseuille Flow |
| 5 PSD | Pore Size Distribution (Analysis) |
| REV | Representative Elementary Volume |
| RTI | Rayleigh-Taylor Instability |

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

## Appendix

**Table 3. Results of the pore size distribution highlighting values for number of pores as well as mean, minimum and maximum pore diameter.**

| | |
|---|---|
| Number of pores n | 1861 |
| Mean pore diameter $\bar{d}_{Pore}$ [µm] | 48.56 |
| Min. porediameter $d_{Pore,min}$ [µm] | 19.09 |
| Max. pore diameter $d_{Pore,max}$ [µm] | 2079.68 |