# Peer review of "Development of a numerical workflow based on $\mu$ -CT-imaging for the determination of capillary pressure-saturation-specific interfacial area relationship in two-phase flow pore-scale porous media systems: A case study on Heletz sandstone"

_Solid Earth, 2016_

## Referee Comment (RC1) · Anonymous Referee #1 · 9 Mar 2016

Authors Peche et al. have submitted the manuscript (ms) entitled "Development of a numerical workflow based on $\mu$-CT-imaging for the determination of capillary pressure-saturation-specific interfacial area relationship in two-phase flow pore-scale porous media systems: A case study on Heletz sandstone" to Solid Earth Discussions.

The authors present a finite element numerical model to describe and simulate fluid-

fluid interfaces using $\mu$-CT obtained pore geometries for multiphase flow problems. The model is applied to a study area of a sandstone environment in Heletz, Israel. It is concluded that this model may be applied to CO2 injection operation planned in the Heletz reservoir, and to apply the new model to other larger areas.

This ms is very well and clearly written. The approach and methodology are thoroughly explained. The model is validated and applied to a real field study. Figures are of good quality. That being said, I would like to see more discussions on prior studies in the Introduction. A one-page intro is not able to adequately set the scene for this new research. The authors should more thoroughly discuss relevant prior studies and more clearly articulate the objective of this ms including the novelty of this ms. For these reasons, which I will outline in the General Comments below, I recommend acceptance of the ms with minor revisions.

General Comments

1. Introduction. The authors almost always point to existing work without giving a brief overview of already existing findings. This should be changed so that the reader understands the knowledge gap that the authors will be filling. The ms is not too long, therefore adding another page or two to the introduction will increase the value of this ms.

2. There are numerous commas missing, which should be corrected. The authors are using punctuation in equations, which should be avoided for clarity. Some equations (e.g. (7)) are not referenced, which should be added. Check all "whereas", which should mostly be "where".

3. Eq. (1). I think there is something wrong in Eq. (1): You are adding nabla u and its transposed form, which does not work. The result is added to pl, which is a matrix. F is a vector, so the expressions on the RHS are all vectors? Is this consistent with the LHS?

[Figure]

4. Eqs. (3) and (4). Please clarify in the text before, which equation does what.

5. P15L13. u is a vector, therefore a "velocity gradient" does not exist. Do you mean velocity difference? How do you calculate that difference when u is a vector?

---

## Referee Comment (RC2) · Anonymous Referee #2 · 16 Mar 2016

Manuscript "Development of a numerical workflow based on $\mu$-CT-imaging for the determination of capillary pressure-saturation-specific interfacial area relationship in two-phase flow pore-scale porous media systems: A case study on Heletz sandstone" by Peche et al. presents a FEM-based methodology for quantification of interfacial area between two immiscible fluids in complex $\mu$-CT obtained geometries in a transient saturation-dependent regime, and implemented for a case of Heletz sandstone.

[Figure]

Conducted modeling of incompressible Navier-Stokes equations in the laminar regime in a pore-scale has been successfully verified. Complexity of the domains used for the modeling gradually increased from idealized single pore geometry to the complex $\mu$-CT-based pore network, defined from a porosity-based REV analysis. Manuscript is clearly written and well organized. Based on presented and discussed modern imaging and computational achievements, proposed methodology has a potential to significantly improve our understanding of the dynamics of a coupled two-phase fluid system with a sharp interface at a micro-scale, followed by a proper upscaling to a macro-scale. Presented methodology would attract an interest of the scientific and industrial communities. The following minor corrections are suggested:

General comments:

I suggest to avoid the multiple abbreviations (PPF, PSD, etc.) explained in the list of abbreviations in the end of manuscript and to leave only the commonly-accepted ones (REV, FEM, etc.)

p.4 bottom: pls change a reference to Comsol (2014) manual to the reference to some handbook or to some other theoretical source.

p6, Model validation: I suggest specifying in the first paragraph that three kinds of the model validations were conducted. Then to specify that first validation addresses single-phase Poiseuille flow, why two others deal with two-phase flow.

p.6, Fig.2: Colorbar addressing the depicted modeled velocities is missing.

p.9 & p.11: Pls insert a sentence with more detailed explanation of REV.

p.15: I suggest changing the title of subsection 6.5 to "Final model setup"

Miscellaneous:

p.5, line 10 "equation is implemented according to equations 3 and 4 " : pls change the wording

p.12, line 14: Pls consider the following substitution: "Model domains are called ->PRESENT? an idealized pore, idealized pore network, $\mu$-CT-based pore and $\mu$-CT-based pore network.", following the increasing level of complexity.

p.15, line 6: a_wn is specified for the first time in this section, so that I suggest calling it by its name.

––––––––––––––––––––––––––

---

## Author Comment (AC1) · 18 Mar 2016

We want to thank the anonymous referee number 1 for his meaningful and valuable comments which indisputable lead to an improvement of the quality of this manuscript. The following section aims at answering to the referee's comments. Answers are given according to the referee's general comment order (which is listed and numbered). This

document is also added as a .pdf with colors and fonts indicating our answers and modified or newly added sections in order to make changes more clear for referee number 1.

1. Introduction. The authors almost always point to existing work without giving a brief overview of already existing findings. This should be changed so that the reader understands the knowledge gap that the authors will be filling. The ms is not too long, therefore adding another page or two to the introduction will increase the value of this ms.

We agree to this point and modified the introduction accordingly. We extended the introduction by including the existing findings from the literature. We consider that the section is now more readable and understandable.

The modified introduction is given in the following section.

"Introduction

Understanding the evolution of the fluid-fluid interfaces in two-phase flow in porous media systems is relevant for a series of engineering and technological applications (e.g., CCS, nuclear waste repository, oil recovery, etc.) (Hassanizadeh and Gray, 1990; Joekar-Niasar et al., 2008; Niessner and Hassanizadeh, 2008; Reeves and Celia, 1996). The quantity of the (typically unknown parameter) interfacial area between two phases restricts kinetic interphase mass and energy transfer (Ahrenholz et al., 2011). In numerical modeling with classical macroscale flow models, these processes are not properly taken into account (Muccino et al, 1998; Ahrenholz et al., 2011). Here, averaged quantities neglecting the interfacial area are often used (Tatomir et al., 2015), which are described to lead to negligence of kinetics of mass transfer and assuming local equilibrium (Niessner and Hassanizadeh, 2009). In order to resolve this, the capillary pressure(pc)-wetting phase saturation(Sw)-interfacial area(awn) relationship using the concept of specific interfacial area per volume described by Hassanizadeh and Gray (1990) can be used.

Based on an extended form of Darcy's law (considering fluid-fluid friction force and interfacial forces), the mathematical fundamentals of pore scale two-phase flow regarding the pc-Sw-awn relationship, are given in Hassanizadeh and Gray (1980; 1990; 1993), and Niessner and Hassanizadeh (2008). The derivation of this porous media specific pc-Sw-awn relationship can be realized using physical experiments in form of e.g. micromodels as described amongst others in Karadimitiou et al. (2013; 2014), Karadimitriou and Hassanizadeh (2012), and Lenormand and Touboul (1988). Corresponding numerical models are given, among others, in Ahrenholz et al. (2011), and Porter et al. (2009). These specific approaches succeed in deriving the pc-Sw-awn relationship, but are based on Lattice-Boltzmann simulations, which are considered as elegant but limited by computational resources (White et al., 2006). Generally, the derivation of this parametric relationship on the pore scale requires a high computational effort as well as a complex numerical model. Arzanfudi and Al-Khoury (2015) present a FEM- based model that enables to simulate CO2 leakage through a wellbore. In that one-dimensional approach, the moving CO2-water interface is successfully traced using the level set method. Amiri and Hamouda (2013) evaluate two approaches for dynamic interface tracking, the level set method and the phase field method, for two-dimensional modeling of two-phase flow with viscosity contrast in dual-permeability porous medium. Both approaches enable to quantify the specific interfacial area between the fluids. They conclude that the phase field method is more successful in complicated porous media, gives more realistic results regarding pressure gradients and fluid profiles and they observed less computation time compared with the level set method.

Several experimental and numerical approaches were developed to measure the interfacial area between the two fluid phases. Current research in the field of CCS focuses on the design and development of field investigation techniques allowing short response times in on-site plume monitoring and detection. Detection and quantification of the CO2-brine fluid-fluid interface could be realized using Kinetic Interface Sensitive (KIS) tracers as described in Schaffer et al. (2013). Several mathematical models for

immiscible two-phase flow and KIS tracer transport in porous media were developed in Tatomir et al. (2013; 2014), and Tatomir et al. (2015) whereas the latter also considers compositional effects. In the latter, a two-phase four component flow and transport model is realized with a kinetic mass transfer of tracers between the two fluids and taking the dissolution of $CO_2$ and brine into account. The pc-Sw-awn relationship is seen as an extension to the standard Brooks-Corey model (Brooks and Corey, 1964). The modeling approaches by Tatomir et al. (2013; 2014) and Tatomir et al. (2015) require the a priori knowledge of the fluid-fluid-solid system specific pc-Sw-awn constitutive relationship (Tatomir et al., 2013).

The present article presents the corresponding model developed for deriving this specific pc-Sw-awn constitutive relationship. In this study, a finite element method (FEM) -based pore-scale model is used in order to determine the dynamic evolution of the $CO_2$-water interface in geometries with gradually increasing level of complexity. Within this case study, we present the workflow that enables the derivation of pc-Sw-awn relationships of sandstone core sample based model domains. We describe a feasible and relatively simple numerical model that enables the dynamic fluid-fluid interface tracking on $\mu$-CT obtained geometries. Initial and boundary conditions as well as spatial and temporal discretization are derived using a novel approach taking simplified idealized geometries based on REV-scale pore geometry statistics into account.

Beside use as input for the model given by Tatomir et al. (2015), the pc-Sw-awn relationships resulting from the described model contribute to desiging the KIS tracer laboratory experiments described in Tatomir et al. (2015). Furthermore, the derived relationship aims at improving the characterization the Heletz sandstone reservoir (Edlmann et al., 2016; Niemi et al., 2016; Tatomir et al., 2016)."

2. There are numerous commas missing, which should be corrected. The authors are using punctuation in equations, which should be avoided for clarity. Some equations (e.g. (7)) are not referenced, which should be added. Check all "whereas", which should mostly be "where".

It is true that we made these mistakes. We carefully rechecked the spelling and punctuation and addressed the issues the referee is addressing to.

3. Eq. (1). I think there is something wrong in Eq. (1): You are adding nabla u and its transposed form, which does not work. The result is added to pI, which is a matrix. F is a vector, so the expressions on the RHS are all vectors? Is this consistent with the LHS?

Yes, we mistakenly formulated a wrong form of the momentum balance of the Navier-Stokes equations (NSE). We particularly thank the reviewer for correcting this mistake and changed the whole momentum balance equation to a form of NSE that is more general (Cengel and Cimbala, 2013).

The corresponding equation is given in the .pdf.

4. Eqs. (3) and (4). Please clarify in the text before, which equation does what.

Yes, we agree. The modified section can be found in the following paragraph.

"The change of phase component concentration over time can be described as the product of diffusion coefficient and chemical potential known as Cahn-Hilliard equation (Cahn and Hilliard, 1958). In the present model, this equation is used in the form of equation (3) using the terms mobility $\gamma$, predefined interface thickness $\varepsilon$, mixing energy density $\lambda$ and a phase field help variable $\Psi$ comprising the chemical potential. "

Corresponding equations are given in the .pdf.

5. P15L13. u is a vector, therefore a "velocity gradient" does not exist. Do you mean velocity difference? How do you calculate that difference when u is a vector?

Term "velocity gradient" was intended to express the difference in velocities at the wall and the velocity which increases with the distance from the wall. Following Reviewer 1's observation the term is changed to "velocity difference".

The comments of anonymous referee number 2 will be addressed soon.

Please also note the supplement to this comment:
http://www.solid-earth-discuss.net/se-2016-39/se-2016-39-AC1-supplement.pdf
* * *

---

## Author Comment (AC2) · 31 Mar 2016

We want to show our gratitude to anonymous referee number 2 for his valuable and reasonable comments. Answers to the comments are listed as given by referee 2 in the following section. This document is also added as a .pdf with colors and fonts indicating our answers and modified or newly added sections in order to make changes

more obvious.

I suggest to avoid the multiple abbreviations (PPF, PSD, etc.) explained in the list of abbreviations in the end of manuscript and to leave only the commonly-accepted ones (REV, FEM, etc.).

We agree and used full terms instead of abbreviations for digital sample approximation, phase field method, plane Poiseuille flow and Rayleigh-Taylor instability.

p.4 bottom: pls change a reference to Comsol (2014) manual to the reference to some handbook or to some other theoretical source.

We agree and changed this reference to another source.

p.6, Model validation: I suggest specifying in the first paragraph that three kinds of the model validations were conducted. Then to specify that first validation addresses single-phase Poiseuille flow, why two others deal with two-phase flow.

We agree that adding such sub-section makes this section more understandable. We added: The model described in this study was validated by reproduction of three analytical solutions. First validation refers to single-phase flow and second and third validation at two-phase flow and fluid-fluid interface tracking.

p.6, Fig.2: Colorbar addressing the depicted modeled velocities is missing.

We understand the point of referee number 2. However, this figure only aims at clarifying the model setup for plane Poiseuille flow (where we only used a subgeometry for the validation). Since only the value at the channel center is important we give this value and only qualitative information about isolines of velocities in the figure description (...blue and red colors define lower and higher velocities respectively.). We believe this is sufficient and a legend specifying velocities is not necessary.

p.9 & p.11: Pls insert a sentence with more detailed explanation of REV.

It is true, a general description of the REV-concept might be very helpful. We added:

Generally, a REV-analysis identifies the minimum porous media volume with similar (e.g.) hydraulic properties of the total porous media volume (Singhal and Gupta, 2010). Further information about the REV can be found in Bear (1988).

p.15: I suggest changing the title of subsection 6.5 to "Final model setup"

We agree and changed the subsection title.

Miscellaneous: p.5, line 10 "equation is implemented according to equations 3 and 4 " : pls change the wording.

This sentence and wording has been changed

p.12, line 14: Pls consider the following substitution: "Model domains are called -> PRESENT? an idealized pore, idealized pore network, micro-CT-based pore and micro-CT-based pore network.", following the increasing level of complexity.

We agree with the referee and changed the sentence to past tense.

p.15, line 6: a_wn is specified for the first time in this section, so that I suggest calling it by its name.

We agree with the referee and substituted awn with interfacial area (awn).

Please also note the supplement to this comment:
http://www.solid-earth-discuss.net/se-2016-39/se-2016-39-AC2-supplement.pdf